# DataMUX: Data Multiplexing for Neural Networks

**Vishvak Murahari**
Department of Computer Science
Princeton University
murahari@princeton.edu

**Carlos E. Jimenez**
Department of Computer Science
Princeton University
carlosej@princeton.edu

**Runzhe Yang**
Department of Computer Science
Princeton University
runzhey@princeton.edu

**Karthik Narasimhan**
Department of Computer Science
Princeton University
karthikn@princeton.edu

## Abstract

In this paper, we introduce *data multiplexing* (DataMUX), a technique that enables deep neural networks to process multiple inputs simultaneously using a single compact representation. DataMUX demonstrates that neural networks are capable of generating accurate predictions over *mixtures* of inputs, resulting in increased inference throughput with minimal extra memory requirements. Our approach uses two key components – 1) a *multiplexing* layer that performs a fixed linear transformation to each input before combining them to create a 'mixed' representation of the same size as a single input, which is then processed by the base network, and 2) a *demultiplexing* layer that converts the base network's output back into independent representations before producing predictions for each input. We show the viability of DataMUX for different architectures (Transformers, and to a much lesser extent MLPs and CNNs) across six different tasks spanning sentence classification, named entity recognition and image classification. For instance, DataMUX for Transformers can multiplex up to 20x/40x inputs, achieving up to 11x/18x increase in inference throughput with absolute performance drops of $< 2\%$ and $< 4\%$ respectively compared to a vanilla Transformer on MNLI, a natural language inference task. We also provide a theoretical construction for multiplexing in self-attention networks and analyze the effect of various design elements in DataMUX. [1]

## 1 Introduction

Deep neural networks are very effective at modeling complex functions from real-valued vector inputs to vector outputs. However, recent studies have hinted that modern networks are vastly overparameterized Kaplan et al. (2020); Allen-Zhu et al. (2019a) and only require a fraction of their parameters for capturing many functions Frankle and Carbin (2018); Frankle et al. (2020). This raises a natural question: *if networks contain greater processing capacity than necessary, could it be possible for them to model a function over multiple inputs simultaneously, similar to how radio channels share bandwidth to carry multiple information streams at once?* Such a capability could enable significant improvements in efficiency and inference throughput without requiring substantial increases in model size or computation.

In this paper, we propose Data Multiplexing (DataMUX), a technique that enables neural networks to process multiple inputs simultaneously as a single "mixed" representation of the same size as a

---

[1]Code is available at https://github.com/princeton-nlp/DataMUX

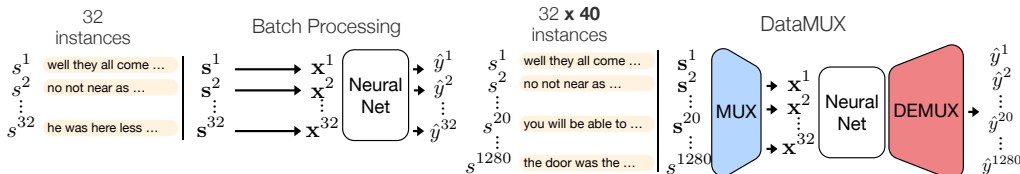

**Figure 1:** Schematic illustrating our proposed DataMUX technique in comparison to traditional minibatch processing in neural networks. Here, DataMUX uses a multiplexing layer (MUX), to multiplex 40 instances into a single representation, passing only 32 inputs to the neural network. The demultiplexing layer (DEMUX) uses the output to generate predictions for all 1280 instances.

single input, during both training and inference (Figure 1). This is complementary to and different from minibatch processing that adds an extra batch dimension to the input, which in effect increases computational and memory requirements. Due to the use of a single combined representation, DataMUX can increase network inference throughput with minimal time and memory overhead.

The key challenge for DataMUX lies in effectively compressing a mixture of signals so that it can be processed by a neural network without sacrificing task performance. To this end, our procedure involves the addition of a multiplexing layer and a demultiplexing layer to both ends of a vanilla neural network. The multiplexing layer applies a fixed linear transformation to each input before combining them to create a single "mixed" representation. This is then fed into the base network, which processes it to obtain a "mixed" vector representation. The demultiplexing layer then converts this vector back into a set of vector representations corresponding to each original input, which is then used to make final predictions for each instance. Our multiplexing and demultiplexing layers are end-to-end differentiable, which allows us to train the entire model jointly through standard gradient descent methods. We find that training time to convergence increases almost linearly with the number of multiplexed in stances as training gets more challenging since the model has to make multiple simultaneous predictions. In order to train multiplexed models effectively, we also introduce a special token retrieval task as a pre-training objective.

We demonstrate the effectiveness of our DataMUX approach using three different architectures (Transformers Vaswani et al. (2017), MLPs, and CNNs) on six tasks spanning sentence classification, named entity recognition and image classification. Our results demonstrate that DataMUX models can successfully process between 2-40 input instances simultaneously, with minimal losses in task performance. For instance, on the sentiment analysis dataset SST-2, our multiplexed Transformer (T-MUX) can handle 40x inputs with only a 2% drop in accuracy compared to a vanilla Transformer. To our best knowledge, this is the first demonstration of large-scale data multiplexing in Transformers. While our results on CNNs and MLPs for image classification are not as strong as those on Transformers, we present them to demonstrate the generality of our method and believe that better multiplexing methods may yet improve them in the future.

We further perform theoretical and empirical analyses to better understand our results. First, we show that it is possible to theoretically construct self-attention networks that can simultaneously process inputs in $N$ independent subspaces. Next, we perform several analyses of various multiplexing strategies and the effect of attention heads and model size. Finally, we also measure throughput statistics to demonstrate the efficiency gains provided by DataMUX (up to 18x inference speedup). We believe multiplexing holds great promise in improving the efficiency and versatility of neural networks and hope that future research can develop improved strategies for DataMUX and expand its applicability to more architectures and tasks.

## 2  Related Work

**Multiplexing in deep learning**  While multiplexing is a standard idea in signal processing Rabiner and Gold (1975), its use in deep learning has been limited. Lu et al. (2020) develop MUXConv, a layer that multiplexes both spatial and channel information for convolutional neural networks (CNNs). Instead of performing *data multiplexing*, these models multiplex different features within the network and only process one input at a time. *mixup* Zhang et al. (2018) proposes a training scheme where networks are trained with a convex combination of instances to predict the instances' label distributions from this combined representation. They find that this scheme provides implicit regularization of models and improves generalization during inference. Unlike our work however,

*mixup* does not preserve the order of the combined instances and therefore mixed instances may not be used during inference. Another line of work uses multiple-input multiple-output training to obtain multiple sub-networks within a neural network for a relatively small number of inputs Havasi et al. (2021); Ramé et al. (2021); Soflaei et al. (2020). These approaches aim to improve single-instance performance through ensembling sub-networks' predictions. In contrast, we are motivated by achieving optimal performance for prediction on multiple inputs and improving model throughput during inference.

**Multiplexing in the brain** On the biological side, several studies hint at the capability of the brain to multiplex and transmit tremendous amounts of information with a single neural code and process them in parallel Blumhagen et al. (2011); Akam and Kullmann (2014); Pirschel and Kretzberg (2016); Hong et al. (2016); Friedrich et al. (2004). Recently, Lankarany et al. (2019) confirmed that neurons in the somatosensory cortex multiplex different types of features (low-contrast, high-contrast) using their timing and rate of synchronous and asynchronous spiking. Rezaei et al. (2021) further provide a computational model to explain multiplexing with an ensemble of homogeneous cortical neurons. These studies increasingly demonstrate that data multiplexing is a key feature of *natural* neural networks.

**Over-paramaterization in neural networks** Our work also relates to the over-parameterization of deep neural networks that has been investigated in several pieces of prior work Allen-Zhu et al. (2019b,a); Radhakrishnan et al. (2020). The lottery ticket hypothesis Frankle and Carbin (2018); Frankle et al. (2020); Malach et al. (2020) demonstrated that only a fraction of neurons in a trained deep model are sufficient to capture a required function. Neural architecture search Zoph and Le (2016); Liu et al. (2018); Elsken et al. (2019) aims to balance performance with parameter efficiency, by discovering the most optimal architectures for each task. There have also been several efforts at leveraging this over-parameterization effect for multi-task learning Hu and Singh (2021). Cheung et al. (2019) propose a method to superpose many models for different tasks into a single neural network, and show that it helps mitigate catastrophic forgetting. Wortsman et al. (2020) is another work along the same lines which uses a base network, along with specialized sub-networks for different tasks. Similar to these paradigms, our work possibly takes advantage of the over-parameterization existing in deep neural networks. However, to our knowledge, we are the first to utilize *data* multiplexing to simultaneously process multiple inputs during both training and inference, and are able to show that a fixed set of parameters can process a mixture of inputs (even up to 40) with minimal loss in accuracy.

## 3 Method

Our primary goal is to process a set of input instances *simultaneously* over a shared neural network (multiplexing) with minimal overhead during inference. To this end, we design DataMUX to consist of three components: a multiplexer module to combine the multiple input instances into a superposed representation, a neural network backbone to process mixed representations, and a demultiplexing module to disentangle the processed representations for individual prediction. A schematic illustrating the flow of representations is shown in Figure 2. We detail the general requirements for each component below, as well as specific implementation details used in our experiments.

### 3.1 Multiplexing

The multiplexer module, denoted $\Phi$, combines a tuple of inputs, either images or sentences from a batch, $(\mathbf{x}^1, \ldots, \mathbf{x}^N)$, for $\mathbf{x}^i \in \mathbb{R}^d$, into a more compact representation $\mathbf{x}^{1:N} \in \mathbb{R}^d$ in an order-dependent way, which enables effective demultiplexing after processing, as well as distinguishing intra-sequence interactions in the case of sequenced inputs (e.g. token sequences). Towards this end, for each input $\mathbf{x}^i$ with index $i \in [1, N]$ of the input tuple, the multiplexer module performs a transformation $\phi^i$ ($\mathbb{R}^d \mapsto \mathbb{R}^d$), on the instance before finally averaging all inputs into a single multiplexed representation as:

$$\mathbf{x}^{1:N} = \Phi(\mathbf{x}^1, \ldots, \mathbf{x}^N) = \frac{1}{N} \sum_{i=1}^{N} \phi^i(\mathbf{x}^i). \tag{1}$$

For sequenced inputs (e.g. token sequences), we combine $N$ sequences by multiplexing token-wise. That is for inputs of the form $\mathbf{x}^i = \{\mathbf{w}_j^i\}_{j \in [1, L]}$, where $\mathbf{w}_j^i \in \mathbb{R}^d$ is a token's input vector representation, this operation uses the same transformation $\phi^i$ for each token in the sequence before

averaging over each position across indices, such that $\mathbf{x}^{1:N} = \{\mathbf{w}_j^{1:N}\}_{j \in [1,L]}$ where $\mathbf{w}_j^{1:N} \in \mathbb{R}^d$ is a multiplexed representation of $N$ tokens at position $j$, i.e., $\mathbf{x}^{1:N} = \{\Phi(\mathbf{w}_j^1, \ldots, \mathbf{w}_j^N)\}_{j \in [1,L]}$.

For $\phi^i$, we experiment with using either (1) a linear projection with a random fixed orthogonal matrix (denoted "Ortho") or (2) the Hadamard product with a fixed Gaussian random vector (denoted "Hadmard", equivalent to a linear map using a diagonal matrix in our case). We hope these transformations map instances at different indices into distinguishable regions and consequently reduce interference between their representations. Finally, this multiplexed representation, $\mathbf{x}^{1:N}$, is used as input to the neural network backbone, which is architecturally unchanged.

### 3.2 Demultiplexing

The output of the neural network backbone will be a multiplexed hidden representation $\mathbf{h}^{1:N}$ of the input $\mathbf{x}^{1:N}$. To make a prediction for each input, one can explicitly disentangle $\mathbf{h}^{1:N}$ into $N$ individual hidden representations, $\mathbf{h}^1, \ldots, \mathbf{h}^N$, respectively. We first obtain each $\mathbf{h}^i$ with a demultiplexing function $\vartheta^i$, i.e.,

$$\mathbf{h}^i = \vartheta^i(\mathbf{h}^{1:N}), \ \forall i \in [1, \ldots, N]. \tag{2}$$

For sequenced input, demultiplexing is done position-wise, i.e., $\mathbf{h}_j^i = \vartheta^i(\mathbf{h}_j^{1:N})$ for each position $j$.

Finally, predictions are made using a shared task head on each inputs' respective individual hidden representation to prevent a substantial increase in the number of parameters and improve training efficiency. We use two alternatives for the demultiplexing function $\vartheta^i$:

**1. MLP Demuxing** This strategy employs $N$ MLPs to generate each indices' hidden representation as $\mathbf{h}^i = \mathrm{MLP}^i(\mathbf{h}^{1:N})$. We use this method for both NLP and vision tasks. Although this method is conceptually simple, it adds learnable parameters proportional to $N$.

**2. Index Embeddings** We generate index embeddings $\mathbf{p}^i$, which are then concatenated to $\mathbf{h}_j^{1:N}$, and transformed by a shared multi-layer network to generate each individual hidden representation, i.e., $\mathbf{h}_j^i = \mathrm{MLP}^{\mathrm{shared}}(\mathbf{h}_j^{1:N}, \mathbf{p}^i)$. To generate the index embeddings $\mathbf{p}^i$, we add a sequence of $N$ special tokens, called the prefix, to the beginning of each sequence of the input tuple. For multiplexing with $N$ sequences, we add $N$ corresponding prefixes, denoted *prefix*$^i$ for $i \in [1, N]$. Each *prefix*$^i$ consists of a index token $\epsilon^i$ in it's $i$'th position while the remaining tokens are a special pad token $\epsilon^{\mathrm{pad}}$. The prefix sequences then take on the following pattern:

$$prefix^1 = [\epsilon^1, \epsilon^{\mathrm{pad}}, \ldots, \epsilon^{\mathrm{pad}}] \quad prefix^2 = [\epsilon^{\mathrm{pad}}, \epsilon^2, \epsilon^{\mathrm{pad}}, \ldots, \epsilon^{\mathrm{pad}}] \cdots \quad prefix^N = [\epsilon^{\mathrm{pad}}, \ldots, \epsilon^{\mathrm{pad}}, \epsilon^N]$$

We then prepend each sequence $x^i$ of the input sequence with the corresponding *prefix*$^i$. The tuple of prepended sequences is then passed to the multiplexing module. When finally generating individual hidden representations, we use the corresponding hidden representation of each index token $\epsilon^i$ as the index embedding $\mathbf{p}^i$.

We use the Index Embeddings ( 3.2) demultiplexing strategy on language tasks for the Transformer architecture. The prefix tokens may implicitly enable the Transformer to do instance-specific computations when processing the multiplexed representation and further enable demultiplexing for large $N$.

### 3.3 Retrieval warm-up for multiplexing

For Transformer models, we find that naively adding the multiplexing and demultiplexing layers to the model fail to converge. This is likely because the gradients for individual instances from the task loss get mixed up in the backward pass. To overcome this, we propose *Retrieval* warm-up – a self-supervised pre-training task to promote the ability of DataMUX models at distinguishing the order and the content of individual sequences in a multiplexed representation. This task consists of retrieving the correct tokens and order for each position and sequence of the input tuple (Figure 4a). Although we could add this loss for every sentence in each position, memory constraints force us to retrieve a token from one random sentence for each token position, yielding the following objective:

$$\mathcal{L}_{\mathrm{Retrieval}}(x^{1:N}) = \sum_j -\log P(w_j^I | \mathbf{h}_j^I), \tag{3}$$

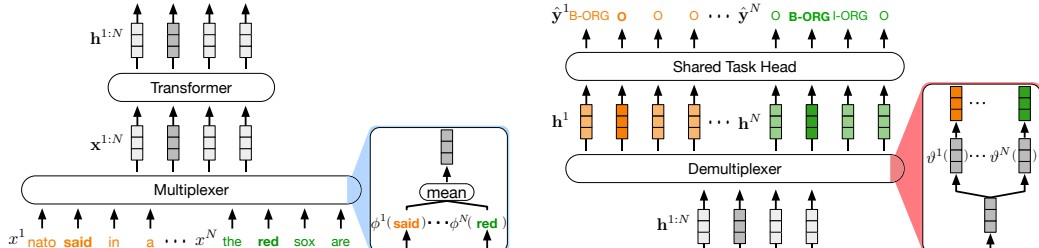

**Figure 2:** DataMUX for Transformers: **(Left)** Given a tuple of $N$ sentences $(x^1, x^2, \ldots, x^N)$, each of length $L$, we first apply a *multiplexing* operation which performs a transformation $\phi^i$ on the embeddings of each sequence $x^i$, such that the same transformation is applied to every token in a sequence $x^i$. The multiplexing operation then aggregates the sequences by averaging over each position, generating a single combined sequence $\mathbf{x}^{1:N} \in \mathbb{R}^{L \times d}$, for embedding size $d$, which will be passed on to the central Transformer model. **(Right)** After processing, we perform a *demultiplexing* operation to the Transformer model's output $\mathbf{h}^{1:N} \in \mathbb{R}^{L \times d}$, to generate hidden representations $\mathbf{h}^1, \mathbf{h}^2, \ldots, \mathbf{h}^N$, corresponding to inputs $x^1, x^2, x^N$ respectively. We finally use these hidden representations to generate predictions for a particular task (e.g. named entity recognition (NER)) using a shared task prediction head.

where $\mathbf{h}_j^I$ is a demultiplexed hidden representation of the $j$-th token in a randomly selected sentence with the index $I \sim \mathcal{U}[1, N]$, generated using the methods described in Section 3.2. We use this objective to optimize all the parameters of the end-to-end multiplexed model. For sequenced inputs, we find the retrieval auxiliary objective crucial to learning and report performance on the retrieval task in Section 4.2.

## 4 Multiplexing for Transformers (T-MUX)

### 4.1 Experimental setup

**Models** We first evaluate the capabilities and limits of data multiplexing specifically for the Transformer architecture on a range of text classification tasks. We apply DataMUX on a 12-layer Transformer based model with a hidden dimension size of 768 and 12 self-attention heads built on the Huggingface Wolf et al. (2019) framework, and refer to the resulting model as T-MUX. We compare our T-MUX models to 2 baselines: **(B1)** A 12-layer 768 hidden dimension vanilla Transformer. **(B2)** A 12-layer 768 hidden dimension Transformer pretrained using the retrieval task described in Section 3.3. Though there is no multiplexing done for **B2** (meaning this operation could be solved by simply copying input tokens to the output layer) we find that the retrieval pre-training produces differences in performance and we show this baseline for completeness.

We also apply DataMUX to smaller models (see **A2**) and compare with similar baselines.

**Tasks** We evaluate our models and the baselines on two types of text classification tasks:

**1. Token-level classification:** This evaluates models' ability to perform token-level tasks on multiplexed inputs. This poses a particular challenge for data-multiplexing models since they must maintain a high level of individual token disentanglement while also producing representations capable of solving the task. We evaluate token-level classification on the CoNLL-2003 Named Entity Recognition (NER) task Sang and Meulder (2003).

**2. Sentence-level classification:** We evaluate models on a subset of the sentence-level classification tasks found in the General Language Understanding Evaluation (GLUE) benchmark Wang et al. (2019): the sentiment classification task SST-2 Socher et al. (2013), the sentence similarity task QQP[2], and the natural language inference tasks MNLI Williams et al. (2018) and QNLI Wang et al. (2019); Rajpurkar et al. (2016). By evaluating on a variety of sentence-level tasks, we can gain a better sense of the capabilities of data multiplexing neural networks on tasks that require aggregating contextual information. Similar to previous works, we prepend a [CLS] token to all sequences and learn a task head on top of the demultiplexed [CLS] token representation.

**Auxiliary retrieval objective** The T-MUX models are all pre-trained using the retrieval warm-up on the Wikitext-103 dataset Merity et al. (2017) as described in Section 3.3. In addition, we also

---

[2]https://data.quora.com/First-Quora-Dataset-Release-Question-Pairs

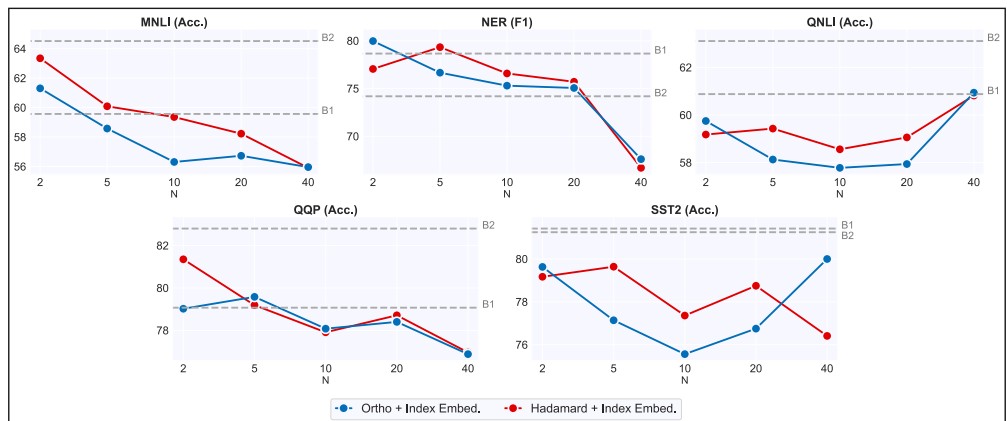

**Figure 3:** Multiplexing task evaluation for T-MUX. Across tasks, we demonstrate multiplexing up to 40 instances without significant drop in performance. Further, we show results here for Index Embedding demultiplexing as the MLP Demuxing method leads to optimization instability for the Transformer architecture. We provide results for the MLP Demuxing method in Appendix A.6.

continue to use the retrieval task as an auxiliary objective during task training. The total loss is a combination of the task loss and retrieval loss (we use $\alpha = 0.1$ in our experiments):

$$\mathcal{L} = (1 - \alpha)\mathcal{L}_{\text{Task}} + \alpha\mathcal{L}_{\text{Retrieval}}, \tag{4}$$

## 4.2 Main results

**(R1) Multiplexing leads to minimal drop in performance even for large** $N$  Figure 3 shows performance across four sentence classification tasks (MNLI, QNLI, QQP, SST2) and a token-level classification task (NER). We observe that it is possible to multiplex up to 40 instances and maintain reasonable performance. For easier tasks like QQP, SST2 and QNLI, we observe that the drop in performance with increasing $N$ is insignificant while for more difficult tasks like MNLI and NER, there is a trade-off between performance and $N$, with performance dropping 10%-15% for 40 instances. Since we've encountered unstable optimization when using MLP Demuxing, we only provide results using Index Embeddings demultiplexing. We find that the multiplexing strategy does not impact performance across different tasks, even slightly increasing for small values of $N$ (e.g. 2, 5). This increase may be attributable to implicit regularization à la *mixup*. We also did not notice much variance between fine-tuning runs and plot variance for the "Hadamard + Index Embedding" setting in Section A.4.

**(R2) Perfect multiplexing on the retrieval warm-up task for large** $N$  Figure 4b shows the test accuracy on the retrieval warm-up task described in Section 3.3. We first note that across different multiplexing and demultiplexing strategies, models have a retrieval accuracy of nearly 100% for up to 20 instances, demonstrating the surprising ability of T-MUX to multiplex perfectly for large $N$. Note that this task does not require any aggregation of context across the sequence and thereby is much easier than sentence classification or token-level classification tasks. Therefore, performance on this warm-up task indicates a soft upper bound on the number of instances we can multiplex for sentence and token-level classification tasks given a particular multiplexing and demultiplexing method.

**(R3) Throughput can be increased multi-fold**  We measure throughput of our multiplexed model (Hadamard + Index Embed) across different number of instances by calculating inference speed for processing ∼20,000 samples on the MNLI dataset. We use four different batch sizes for all the configurations and take the max throughput (details in A.8). Figure 4c shows that multiplexing increases throughput many folds (18x for 40 instances, 11x for 20 instances). T-MUX enables superior throughput as batch size can be effectively increased by a factor of $N$. We would expect the speedup to scale linearly with $N$, however a large $N$ corresponds to more prefix tokens which increases the sequence length. Therefore, having 40 instances leads to almost a 20x speedup as opposed to the expected 40x. Future work can potentially improve this speedup by designing better multiplexing and demultiplexing strategies.

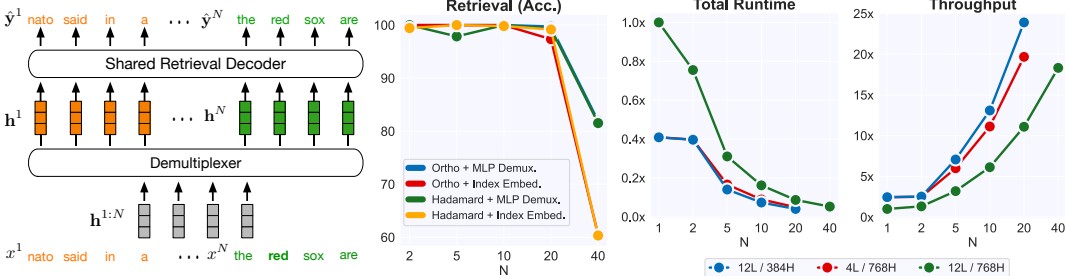

Figure 4: (Left a) Given the Transformer's combined output representation $\mathbf{h}^{1:N}$ generated from the input $(x^1, x^2, \ldots, x^N)$, the goal of the retrieval task is to retrieve the original tokens from $(x^1, x^2, \ldots, x^N)$. Since $\mathbf{h}^{1:N}$ is order-dependent, retrieval requires distinguishing not just which tokens were originally input to a position $j \in [1, L]$, but also which sentence $i \in [1, N]$ each belonged to. (Center b) Accuracy for the retrieval warm-up task. Surprisingly, models are able to retrieve words with $100\%$ accuracy up to 20 instances across most multiplexing and demultiplexing strategies. (Right c) Runtime and throughput numbers of T-MUX models on 20K MNLI instances, normalized by a base model's performance without multiplexing. T-MUX (12L/768H) can multiplex up to 40 instances leading to $\sim 18$x speedup.

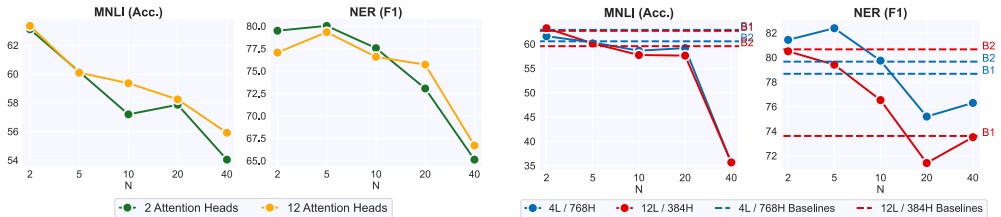

Figure 5: (Left a) Understanding the role of attention heads in multiplexing. Reducing attention heads to 2 does not impact performance greatly, suggesting that attention heads are not core to multiplexing. (Right b) Multiplexing performance with smaller model capacities. Smaller models can multiplex up to 20 instances without significant drop in performance.

## 4.3 Analysis

**(A1) The number of attention heads seems invariant to multiplexing** To understand the role of the number of attention heads in multiplexing, we train a variant of T-MUX with 2 self-attention heads per layer. We use T-MUX with the (Hadamard + Index Embed.) configuration. We find that reducing self-attention heads has minimal effect on performance on the retrieval warm-up task and achieves a retrieval accuracy of $\sim 100\%$ up to $N = 20$. We then find that on token and sentence-level classification tasks, T-MUX with 2 self-attention heads performs comparably to T-MUX with 12 self-attention heads (Figure 5a).

**(A2) DataMUX also provides throughput boost with smaller Transformers** We further investigate the applicability of DataMUX to smaller Transformer models. We choose two smaller Transformers[3] to multiplex: a 12 layer with hidden size of 384 (12L / 384H), a 4 layer with hidden size of 768 (4L / 768H). Figure 5b shows that these smaller T-MUX models can also multiplex up to 20 instances with competitive performance. Figure 4c illustrates the speedup from the smaller models. As the smaller models can only multiplex up to 20 instances with reasonable performance, we see that multiplexing with 20 instances provides an even higher throughput of 25x, compared to only 18x for the full-sized T-MUX with 40 instances.

**(A3) Performance varies more across different indices as $N$ increases** The prediction of an instance is conditioned on the index of the instance. Figure 7b illustrates performance on MNLI for different indices across different choices of $N$ for the (Hadamard + Index Embed.) configuration. We observe that performance varies more across different indices for large $N$ (For $N = 40$, performance varies across $\sim 10$ percentage points).

**(A4) The demultiplexed representation of an instance is robust to the other instances it is multiplexed with** To analyze whether the representation of an instance changes with respect to the other samples it is multiplexed with, we randomly select 10 instances $x_1, x_2, \cdots x_{10}$ from the

---

[3]We performed an empirical study to determine models that performed best, see A.7 for details.

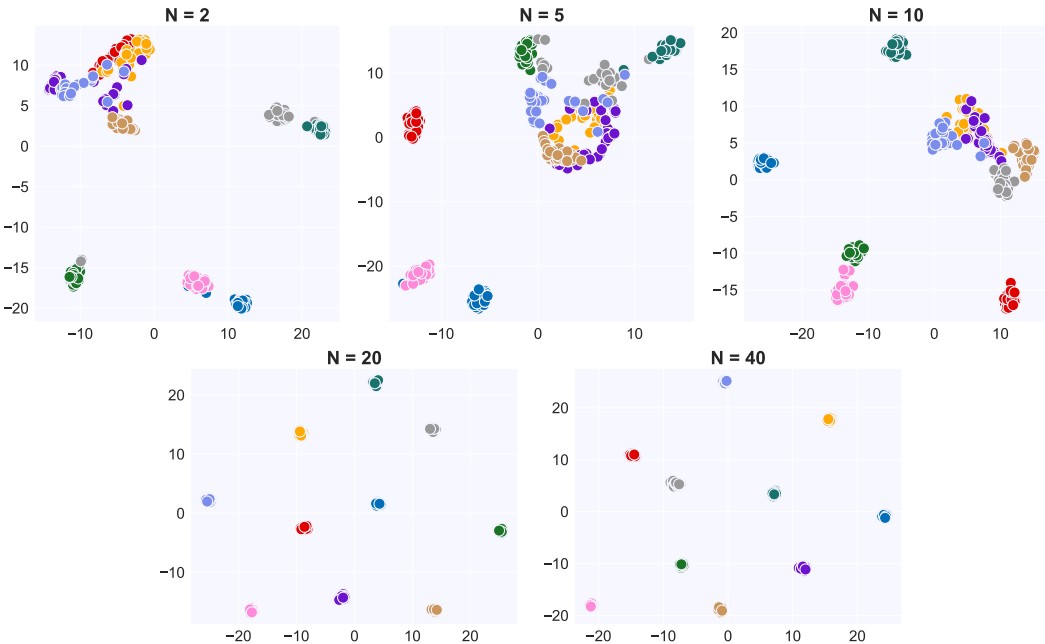

**Figure 6:** T-SNE plots to understand how the demultiplexed representation of an instance changes with respect to the set of instances it is multiplexed with. Each color (10 in total) a single input sequence. Across different $N$, we find that the demultiplexed representation of an instance is not significantly impacted by the set of instances it is multiplexed with.

MNLI dataset and multiplex each of them (separately) with 30 different sets of other instances $y_1^i, y_2^i, \cdots, y_{N-1}^i$ for $i \in [1 \cdots 30]$. This generates 30 different demultiplexed representations for each of the 10 selected samples. We then visualize the resulting points by reducing the 768 dimensional demultiplexed representation to a 200 dimensional vector with PCA ( F.R.S. (1901)), followed by t-SNE ( van der Maaten and Hinton (2008)) visualization. Figure 6 shows these clusters for different values of $N$. We find that across different values of $N$, all the points corresponding to the same $x_i$ are very close to each other, which suggests that the representation of an instance is not significantly influenced by the set of instances it is multiplexed with.

## 4.4 Theoretical construction for multiplexing in self-attention models

To provide some theoretical explanation for why Transformers may be amenable to multiplexing, we also detail below a construction for self-attention based neural networks which process multiplexed token embeddings in $N$ independent subspaces. Our construction relies on particular structural properties of singular spaces of linear transformations across layers – we provide a brief sketch below, with more details in Appendix A.3.

In a model with self-attention, the multi-head attention projects the queries, keys, and values $h$ times with different, learned linear projections to $d_K$, $d_K$, and $d_V$ dimensions respectively. Thus, given a sequence of $d$-dimensional multiplexed token embeddings $\{\mathbf{w}_t^{1:N}\}_{t \in [1,L]}$, each head looks like:

$$head_i(t) := \sum_{t'=1}^{L} \left[ \frac{\exp\left(\frac{(\boldsymbol{W}_i^K \mathbf{w}_{t'}^{1:N})^\top \boldsymbol{W}_i^Q \mathbf{w}_t^{1:N}}{\sqrt{d_K}}\right)}{\sum_{t''} \exp\left(\frac{(\boldsymbol{W}_i^K \mathbf{w}_{t''}^{1:N})^\top \boldsymbol{W}_i^Q \boldsymbol{u}_t}{\sqrt{d_K}}\right)} \boldsymbol{W}_i^V \mathbf{w}_{t'}^{1:N} \right]. \tag{5}$$

Now, $\mathbf{w}_t^{1:N} = \frac{1}{N} \sum_{k=1}^{N} \phi^k(\mathbf{w}_t^k)$, and let us assume each function $\phi^k$ can be trained to project each embedding into a subspace that is least linearly-dependent with the others. So, if we define $\boldsymbol{u}_t^{(k)} := \frac{1}{N} \phi^k(\mathbf{w}_t^k)$, we assume $\langle \boldsymbol{u}_t^{(k)}, \boldsymbol{u}_{t'}^{(k')} \rangle \approx 0$ for all pairs of indices $k \neq k'$ and all positions $t$. To preserve this independent subspace structure after self-attention, we will first need

$$\langle \boldsymbol{W}_i^V \boldsymbol{u}_t^{(k)}, \boldsymbol{W}_i^V \boldsymbol{u}_t^{(k')} \rangle = \boldsymbol{u}_t^{(k)\top} \left( \boldsymbol{W}_i^{V\top} \boldsymbol{W}_i^V \right) \boldsymbol{u}_t^{(k')} \approx 0. \tag{6}$$

This is achievable if the eigenvectors of $\boldsymbol{W}_i^{V\top}\boldsymbol{W}_i^V$ can be grouped into $N$ non-overlapping subsets $\{\boldsymbol{r}_1^{(1)}, \ldots, \boldsymbol{r}_m^{(1)}\}, \cdots, \{\boldsymbol{r}_1^{(N)}, \ldots, \boldsymbol{r}_m^{(N)}\}$, where $\boldsymbol{r}_\ell^{(k)}$ are orthonormal vectors (since the Gramian is real symmetric), and span the same input subspaces, denoted as $\mathcal{D}^1, \ldots, \mathcal{D}^N$. In this case, the vector after transformed by $\boldsymbol{W}_i^V$ can be expressed as a sum of $N$ vectors $\boldsymbol{v}_{i,t}^{(1)}, \ldots, \boldsymbol{v}_{i,t}^{(N)}$ in dual subspaces $\mathcal{D}_V^1, \ldots, \mathcal{D}_V^N$ which are independent of each other. The linear maps from $\mathcal{D}^k$ to $\mathcal{D}_V^k$ still allows rich operation on each component of a multiplex input without interference by other components.

In addition to decompress-able value vectors, we can set the query and key matrices, $\boldsymbol{W}_i^Q$ and $\boldsymbol{W}_i^K$, to have some subsets of right and left singular vectors such that span $\mathcal{D}^1, \ldots, \mathcal{D}^N$ and $\mathcal{D}_V^1, \ldots, \mathcal{D}_V^N$. Then, we can show that the inner product of the query and keys of the $i$-th head can be rewritten as:

$$\left(\boldsymbol{W}_i^K \mathbf{w}_{t'}^{1:N}\right)^\top \boldsymbol{W}_i^Q \mathbf{w}_t^{1:N} = \sum_{k=1}^N \tau_{i,t,t'}^{(k)}, \tag{7}$$

where $\tau_{i,t,t'}^{(k)}$ is a scalar only depending on the $k$-th input sequence. Thus, the self-attention operation at each position can be seen as retrieving values based on the average of query-key similarity scores of $N$ sequences.

$$head_i(t) := \sum_{t'} \left[ \frac{\exp\left(\sum_k \tau_{i,t,t'}^{(k)}/\sqrt{d_K}\right)}{\sum_{t''} \exp\left(\sum_k \tau_{i,t,t''}^{(k)}/\sqrt{d_K}\right)} \sum_k \boldsymbol{v}_{i,t}^{(k)} \right] \tag{8}$$

This average retrieval using soft-max could be a desired property as implicit regularization. However, if we want perfect non-interference in retrieval, the network always has an option to specialize each head to only focus on one input sequence, by setting $\tau_{i,t,t'}^{(k')} = 0$ for all $k' \neq k$, which is easily achievable by controlling singular values of $\boldsymbol{W}_i^Q$ or $\boldsymbol{W}_i^K$. More details of this construction are provided in Appendix A.3. Figure 4.3 suggests that number of attention heads do not significantly impact our empirical multiplexing results. Therefore our theoretical construction does not fully explain what models learn empirically but nonetheless provides an existence proof for multiplexing under certain assumptions.

## 5 Multiplexing for MLPs and CNNs

We also investigate multiplexing for multilayer perceptrons (MLPs) and convolutional neural networks (CNNs). Certain cases of data multiplexing have been explored for convolutional architectures on image classification as techniques for robustness and data augmentation Ramé et al. (2021); Havasi et al. (2021); Soflaei et al. (2020). These works implicitly employed the frontend layers of a convolutional net as multiplexing layers and suggest that a convolutional neural network can learn at most 3-4 independent subnetworks concurrently Havasi et al. (2021). Since Transformers can be multiplexed for up to 40 instances without a severe performance drop, we explore the DataMUX scheme for MLPs and CNNs on the MNIST image classification task LeCun and Cortes (2005).[4] While our results for these architectures are not as strongly positive as for T-MUX, we believe that future work on better multiplexing strategies can make them more viable.

**Multiplexing for MLPs** Figure 7a (solid lines) illustrates performance of DataMUX for MLPs with various multiplexing strategies, paired with the MLP demultiplexing strategy. As a baseline, we show multiplexing using the identity transformation before combining instances (MLP baseline). Since this transformation does not preserve the order of the multiplexed instances, accuracy expectedly decreases on the order of $1/N$. Multiplexing with random orthogonal matrices (MLP + Ortho) works for up to 8 instances with an accuracy of $\sim 78\%$, compared to $\sim 95\%$ for the baseline without multiplexing. Since random orthogonal projections cannot separate inputs perfectly, we also experiment with a set of $N$ low-rank independent transformations for multiplexing (MLP + LowRank) and observe that this improves performance slightly for larger $N$ (by $5\%$ for $N$=8).

**Multiplexing for CNNs** Results for CNN multiplexing are shown in Figure 7a (dashed lines). Like in the MLP case, all methods shown use the MLP demultiplexing strategy. We notice that the baselines (CNN baseline) has a roughly $N$-fold performance drop because of the unidentifiability of input order. The "Ortho" transformation performs quite poorly for CNNs (only $\sim 56\%$ for $N = 8$).

---

[4]Implementation details can be found in Appendix A.10

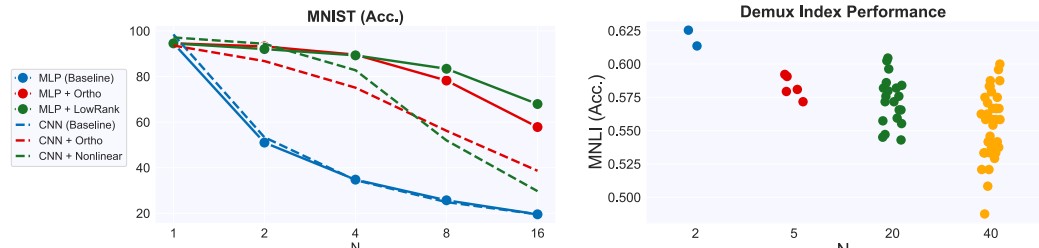

**Figure 7: (Left a)** Average test accuracy for multiplexing MLP and CNN for $N = \{1, 2, 4, 8, 16\}$ on the MNIST classification task for different multiplexing strategies. MLPs can be multiplexed up to $N = 8$ without significant drop in performance while CNNs can only be multiplexed up to 4 instances. **(Right b)** Performance results for different demultiplexing output indices. Results are reported for a Transformer model with the Hadamard product multiplexing and Index Embeddings demultiplexing. Variance of performance across different indices with increasing $N$.

This is likely because this transformation destroys the property of spatial locality that CNNs rely on. We therefore also explore $N$ two-layer convolutional networks with a *tanh* activation as out multiplexing transformations $\phi^i$ (CNN + Nonlinear). We note that even with this transformation, the dimensionality of the multiplexed representation still remains equal to the dimensionality of a single input. We find that for $N \leq 4$, performance is above $80\%$, which is significantly better than CNN+Ortho, but performance drops rapidly for $N > 4$. We show results of other multiplexing strategies for CNN in Appendix A.11.

Overall, multiplexing for MLPs and CNNs seems to be more challenging than Transformers as evidenced by the sharper performance drops with increasing $N$. However, we believe these numbers are still non-trivial and show the potential for multiplexing in CNNs and MLPs, with multiplexing and demultiplexing strategies better suited for these architectures.

## 6 Discussion

In this work, we have shown that neural networks can be trained to multiplex, i.e. process multiple inputs using a single "mixed" representation. We introduced DataMUX, a novel setting for training and inference with multiplexing, and demonstrate its viability using a variety of network architectures and tasks. Our results reveal the surprising ability of neural networks to predict on up to $40$ input instances simultaneously using data multiplexing. Our efforts have sought to show the potential for data multiplexing using methods that require limited computational overhead or added learned parameters; achieving results that incur minimal degradation in performance while simultaneously increasing system throughput dramatically.

While the scope of this paper is to demonstrate the general ability and potential for neural networks to use data multiplexing, there are several avenues for future research to investigate the limits and theoretical implications of DataMUX. Directions of particular interest include: large-scale pre-training, multi-modal processing, multi-lingual models, different mechanisms of multiplexing and demultiplexing, and a more rigorous understanding of the architectural, data, and training conditions enabling data multiplexing.

## 7 Acknowledgements

We thank Ameet Deshpande, Shunyu Yao, Jens Tuyls, Tianyu Gao and Mengzhou Xia for their valuable feedback on early drafts and encouragement throughout the course of this project, and the anonymous reviewers for their suggestions on improving the paper.

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
