# OpenReview forum: "DataMUX: Data Multiplexing for Neural Networks"
_NeurIPS.cc/2022/Conference — NeurIPS 2022 Accept_

### Official Review · Reviewer_N4gg · 2022-07-06

**Rating:** 7
**Confidence:** 4
**Soundness:** 2 fair
**Presentation:** 3 good
**Contribution:** 3 good

**Summary:**

The authors propose to train a single encoder (mainly Transformer, but also CNN, MLP) to process multiple samples in parallel with the goal of approximating batch-processing without the computational overhead added by default batchwise processing. To this end, they first mix the input samples together (and add order tokens), then apply the encoder, implicitly forcing it to learn a form of parallel processing of this joint representation, and then de-mix the representation to obtain the prediction for each individual sample. Experiments indicate that Transformers may be flexible enough to process up to 20 samples in parallel (giving a 11x speed increase) while only losing 2-6pp.
performance.


**Questions:**

- If I understand correctly, the goal is that the processing of each input should not influence each other input. Does this hold theoretically (e.g., decompositional Jacobians) or experimentally (e.g., resampling inputs and recording differences in the output)?
More drastically formulated, how much can the prediction of one sample be influenced by the others in its batch? Does this open an ethically concernable weakness to attacks?
- In this paper, the architectures (Transformers, CNNs) are viewed from a universal approximator "black-box" perspective. Which inductive biases implemented by these architectures motivate this claim, and which threaten to deteriorate the desired independence?
- The model is trained on a fixed batch size. Can we hand over an incomplete batch to the model, e.g., by zero-padding or repeating the input (like we can for normal batch-processing)?
- The two plots in Figure 4c show the same, only with inverted axes. I would find it more useful to use the space for additional information, such as increase in memory complexity (which was claimed to "not rise substantially"). Maybe consider a Pareto chart. This would also allow showing several baselines, i.e., B1 and B2 transformers with several batch sizes
- I do not quite understand the Retrieval warm up: Why do we want to predict the whole input representations from the output, as opposed to just their order token?
- Please provide a reference to the Index Embeddings Demultiplexing strategy in line 157.
- A minor cosmetic critique is that plots are referenced to as, e.g., 4a, 4b, while the plots themselves are not labeled with a and b
- Will code to reproduce the experiments be made available upon acceptance?


**Limitations:**

- Most technical and experimental limitations were adressed humbly and openly (see strength 5)
- The paper opens a field with diverse positive and negative ethical implications which remained unadressed (see weakness 6)


**Strengths And Weaknesses:**

Strengths:
- I see the main value of this work in formulating a useful, novel task and taking a first shot at it (conditional on it being a new task, which I cannot confidently ascertain since it is not my primary
field)
- If the model is able to guarantee independent processing of the inputs, this might have impact on causality and disentanglement literature, as a form of independent mechanisms
- The main hypothesis (efficiency increase without big drops in
performance) is experimentally well investigated for several Transformers and language tasks
- The authors also introduce small methodological novelties (e.g., retrieval warm-up)
- Fluently written and error-free language, in particular very informative headings, and honest about how broad or limited the claims and evidence are


Weaknesses:
- The important side hypothesis (inputs should not influence each
other) is only speculated about ("we hope these transformations [...] reduce interference", "may implicitly enable [...] instance-specific computations"), with only limited experimental (toy case in 4.2(R2)) and theoretical support (4.4 argues that the family of functions a Transformer comprises contains some with independent processing, but not whether these are actually picked during training) (see also question 1)
- Another side claim is that vanilla Transformers with high batchsize (i.e., achieving high throughput) are limited due to their memory consumption, but the memory consumption overhead of the proposed method is not measured and compared to them (see also question 4)
- The experiments for MLPs and CNNs on MNIST are very simplistic. I appreciate them as outlook, but would appreciate being more humble with them in abstract (like, e.g., done in the introduction)
- The results presented in the abstract appear cherrypicked since the weaker B1 was chosen as baseline
- The experiments appear not to be repeated on several random seeds
- The paper opens a novel field with both ethical chances (reduction of computational consumption) and risks (attacks due to confounded processing), and would benefit strongly from discussing them

---

> ### Author Response · Authors · 2022-08-02
> **R4 Response**
>
> **Interference among multiplexed instances**: We do provide theoretical guarantees about interference between different multiplexed instances. However, our empirical results indicate that the interference between different multiplexed instances is minimal. Firstly, section A.4 in appendix indicates that the performance does not vary significantly with different seeds. This implies that the set of samples which are multiplexed has little bearing on the task performance. Moreover, our empirical results also suggest lack of significant interference as effective multiplexing would not be possible with significant interference.
>
> **Moreover, based on your suggestion, we have added a more thorough empirical analysis in A.12**. To understand how the representation of an instance changes with respect to the other samples it is multiplexed with, we randomly select $10$ ‘anchor’ inputs $x_{1}, x_{2}, \cdots x_{10}$ from the MNLI dataset and multiplex that ‘anchor’ input with $30$ different sets of instances $y_{1}^i, y_{2}^i, \cdots, y_{N-1}^i \text{ for } i \in [1 \cdots 30]$, for various N values. We then visualize the resulting demultiplexed representations by applying dimension reduction techniques PCA + t-SNE. Our plots show neat, separated clusters of each instance indicating that the demultiplexed representation of an input is not significantly affected by which set of other instances it is multiplexed with.
> We agree with you that a more thorough exploration (e.g. using saliency or other similar techniques) could provide more deeper insights into this phenomenon and would be a great direction for future work! We have also beefed up discussion on implications for attacks/privacy in the “Ethical considerations” section (A.1)
>
> **Fixed batch size**: It is possible to extend a batch by simply adding dummy inputs (by sampling a few instances from the training corpus) and ignoring the outputs of the dummy inputs. However, zero-padding might not work well as multiplexing with a zero tensor may lead to an out-of-distribution input to the model.
>
> **Memory overhead of multiplexing**:  Thank you for bringing this up. Based on your suggestion, we also analyze the memory overhead of our approach is A.13. We find that the GPU memory of our computational graph does increase linearly with the number of instances. However, the rate of increase is very small.
>
> **Softening claims about CNNs in the abstract**: We have updated the abstract to say “a much lesser extent for CNNs and MLPs ”.
>
> **Results in abstract**: We have updated to clarify this better and be consistent with Introduction as suggested.
>
> **Variance of results across different random seeds**: Please refer to A.4. We find that performance does not vary significantly with different seeds.
>
> **Ethical risks and lack of discussion**: Thank you for bringing this up. Interference between different multiplexed instances could potentially have privacy implications. We have updated the “Ethical considerations” section (A.1) with pertinent discussion.
>
> **Inductive biases which enable multiplexing**: Our results indicate that multiplexing is more effective for Transformers and MLPs than CNNs. This suggests that the spatial locality assumption in CNNs might make multiplexing more trickier to learn for CNNs as the spatial locality assumption might not hold after applying the multiplexing function. This is also suggested by our experiments in section A.10, where we present alternative multiplexing strategies for CNNs. The results clearly indicate that multiplexing functions which try to preserve the spatial locality assumption perform much better than their counterparts. We believe future work in improving multiplexing functions for CNNs could lead to more effective multiplexing.
>
> **Retrieval warmup clarification**: The retrieval warmup task is an auxiliary self-supervised task to prime the model for multiplexing. Since we want to encourage the model to be able to accurately preserve the semantic representation of each representation through multiplexing, we predict the input representations. Predicting just the order is an easier task that does not require preserving the semantics of the input, and can easily be done by just copying the prefix token during demultiplexing. In other words, the retrieval warmup teaches the model to learn MUX and DEMUX with minimal loss of information in the forward pass.
>
> **Availability of Code**: Yes, we will publicly release the code upon acceptance. We have also attached our code in the supplementary material.
>
> **Reference for index embeddings on line 157, referring to subfigures with a. and b.**:  Thank you for pointing out these inconsistencies! We have revised the draft accordingly.

---

### Official Review · Reviewer_BEtA · 2022-07-07

**Rating:** 7
**Confidence:** 3
**Soundness:** 3 good
**Presentation:** 3 good
**Contribution:** 3 good

**Summary:**

This paper proposed DataMUX, a method to allow DL models to process multiple inputs at the same time using a single representation. They conduct their experiments on token and sentence level classification for Transformer, MNIST for CNN and MLPs.

**Questions:**

1. From the experiment, it seems like a single representation can actually contain much information from multiple inputs, which is even enough to decompress it(demultiplexing). So is it possible that a traditional Bert [CLS] token can achieve the same thing with multiplexing pretraining?
2. From section 4.4, even though the self-attention-based models are suitable for multiplexing in certain cases, it seems like it still requires multiplexing pretraining to reach the desired parameters. Therefore, I guess multiplexing and demultiplexing layers cannot just be added to any type of common language model right(e.g. Bert, Roberta)?

**Limitations:**

The authors addressed the limitations and potential negative impact of their work in the appendix.

**Strengths And Weaknesses:**

Strengths:
  1. The motivation here is sufficient, especially the connection to biological perspectives.
  2. Their experiment results show data multiplexing can speed up the throughput while maintaining performance.
  3. It is great to have a theoretical explanation of the proposed method.

Weaknesses:

  1. Please refer to the questions

I would recommend this paper be accepted. Overall, I think this paper has solid motivation from both theoretical and intuitive perspectives. The authors conducted many experiments to show that DataMUX is effective in many tasks. I think there is a lot to explore with the connection between over-parameterization and multiplexing, too, that would be interesting to other researchers.

---

> ### Author Response · Authors · 2022-08-02
> **R3 Response**
>
> **Traditional BERT CLS token and multiplexing**: For all our sentence classification experiments, we prepend a [CLS] token to all sequences and learn a task head on top of the demultiplexed [CLS] token representation. We describe this in greater detail in section 4.1.
>
> **Multiplexing/Demultiplexing layers for pretrained language models**:  We did try to add multiplexing layers to RoBERTa weights but were unable to get positive results just by finetuning the model. We think the key issue here is that the data distribution seen by a pretrained language model is very different from that of a multiplexed model as the multiplexing operation changes the input distribution significantly. Hence, getting the pre-trained LM to move out of its local optimum to find a new optimum for the multiplexed inputs is not trivial. We do however think that it might be possible to more cleverly align the two distributions, which could be a great topic for future work.

---

> > ### Comment · Reviewer_BEtA · 2022-08-06
> > **For the response**
> >
> > Thank you for the response.

---

### Official Review · Reviewer_RBia · 2022-07-11

**Rating:** 8
**Confidence:** 5
**Soundness:** 3 good
**Presentation:** 4 excellent
**Contribution:** 4 excellent

**Summary:**

The authors explore the idea of data multiplexing to accelerate both training and inferencing of neural networks. A data multiplexer is added at the input of the network to multiplex or "mix" N samples into a single one. The network processes this single sample, and then the final answer is demultiplexed near the output of the network back to N answers. Potentially, the method can achieve a theoretical speedup of N times over the original network. The multiplexer and demultiplexer are differentiable and hence trainable. The multiplexer applies a linear transform to each sample and averages per position in a sequence. The demultiplexer is applied for each position to recover the N sequences. During training gradients seem to be superposed or mixed up, failing to converge for Transformers. An auxiliary loss is proposed to overcome this issue. For transformers, the authors share an explanation why is possible to mutiplex the data. In the experimental section, the idea is evaluated on NER, sentiment analysis, sentence similarity, and NLI. The models include Transformers, CNNs and MLPs. The results show typically a drop in performance when N increases.



**Questions:**

* What is the name of the model used from the Huggingface hub?
* What does cause in some cases that for increasing N, the model improves in performance (e.g., SST-2 N=40 for Ortho+Index Emb)?
* What is the original performance of the model (i.e., N=1) in each plot in Figure 3?

**Limitations:**

To the best of my understanding, limitations have been addressed.

**Strengths And Weaknesses:**

The idea of data multiplexing is not new (as pointed out in the text), the application to deep neural networks seems novel. This leads to a very interesting work presented in the paper. This work setups the ground for further improvements and novel methods in this community as well. In addition, I appreciate the practicality of the method, including the discussion on the transformers and the analytical analysis on their properties for exploiting multiplexing. The experiments are convincing and show the limitations when N is large, suggesting there is room for improvement.

The paper is nicely written and clear, as well as supported by nice diagrams.

---

> ### Author Response · Authors · 2022-08-02
> **R2 Response**
>
> **Hugging face hub model name**: We use a Transformer-base model from Hugging face and you can find the configs in the code released in our supplementary material.
>
> **Performance increases in some cases for large N**: This is a good question, and one that we also were intrigued about. A related work mixup (Zhang et. al 2018) suggests that training a network by inputting a convex combination of inputs and predicting a convex combination of labels has a regularizing effect. So, one potential explanation could be that training networks with multiplexing could similarly have a regularization effect for easy tasks like SST-2.
>
> **Original performance of the model (i.e., N=1)**: Exact numbers for B1 and B2 in Fig. 3 are:
> |    | mnli  | ner   | qnli  | qqp   | sst2  |
> |----|-------|-------|-------|-------|-------|
> | B1 | 59.57 | 78.66 | 60.88 | 79.07 | 81.42 |
> | B2 | 64.52 | 74.19 | 63.11 |  82.8 | 81.25 |

---

> > ### Comment · Reviewer_RBia · 2022-08-05
> > **Re: R2 Response**
> >
> > Thank you for your response.

---

### Official Review · Reviewer_DLX3 · 2022-07-14

**Rating:** 7
**Confidence:** 4
**Soundness:** 3 good
**Presentation:** 4 excellent
**Contribution:** 3 good

**Summary:**

The basic idea is as follows: multiplex multiple inputs into one input of the same size, then feed the model's output into a module at the end that untangles the multiplexed representations and makes a prediction for each input individually.

The multiplexing module randomly projects inputs then does an element-wise average. Demultiplexing is achieved by feeding the outputs into a demultiplexing function, which is either a set of N MLPs (where N = number of multiplexed inputs), or by prepending each input sequence with an index token such that a single demultiplexing head can use the index embedding to distinguish between inputs. To work with sequenced inputs, multiplexed models need to be pretrained on a self-supervised “retrieval” task which requires selecting the correct token for each possible position across the set of input sequences. The authors primarily evaluate their method on transformers and NLP tasks, and find that multiplexing results in modest accuracy drops but large throughput increases.

**Questions:**

Overparameterization/unused model capacity are presented as motivations for the development of and reason for success of the method. However, I don't think the analyses in the paper justify this sufficiently. First, the results for the small and baseline models should be compared directly. The following experiments would more thoroughly answer this question: Reduce the model size until performance begins to decline substantially. If model capacity is why multiplexing "works", then the multiplexing models' performance should decline sooner than the non-multiplexing models' performance. The same result should be observed when pruning model weights: performance as a function of weights pruned should be more favorable for non-multiplexed models than for multiplexed models.

The theoretical examination of multiplexing is interesting. I encourage the authors to test these theoretical predictions: for example, do the singular values of the query and key matrices differ between the baseline and retrieval-pretrained networks in the manner predicted by the theoretical examination? Furthermore, if reducing the number of attention heads has little effect on multiplexing capability, does it even make sense to focus the theory on the role of attention heads in multiplexing?

The authors demonstrate that pretraining is necessary for learnable multiplexing, and that retrieval pretraining can affect downstream task performance, but they don’t demonstrate that retrieval pretraining per-se is necessary for learnable multiplexing. While I suspect it is, I think it would be useful to show that other types of pretraining do not affect the learnability of multiplexing.

This method is presented as providing a resource savings due to the inference speedup. However, assessing the resource savings is difficult for a number of reasons. First, the pretraining protocol does not seem to be discussed at all other than a description of the task. Second, multiplexing causes an accuracy decrease relative to baseline models. Is it possible to compensate for this decrease by e.g. training longer? It would be worthwhile to see accuracy vs. training time plots for the different training configurations.

Is it justified to compute speedups for small transformers relative to the baseline size transformer? This confounds the effect of transformer size on speedup with the effect of the multiplexing in small models on speedup.

Is it possible to learn the multiplex projection?


**Limitations:**

The authors do a satisfactory job addressing the limitations of their work.

**Strengths And Weaknesses:**

Strengths: A clever method that I am somewhat surprised the authors were able to make work. The results are clearly presented. The inference speedups appear meaningful.

Weaknesses: I'm not sure how relevant the theory is. Some of the stated motivations are not very thoroughly addressed by the analyses. Total resource savings is unclear.

I think the paper is acceptable in its current form, and I think it would be excellent if revised appropriately.

---

> ### Author Response · Authors · 2022-08-02
> **R1 Response**
>
> **Resource savings and memory overhead**:
> Based on your suggestion, we have added an analysis and discussion of multiplexing vs memory overhead in section A.13. We find that the GPU memory of our computational graph does increase linearly with the number of instances. However, the rate of increase is very small (4x memory usage for N=40 multiplexing).
>
> **Overparameterization hypothesis**. As already mentioned in the paper, the overparameterization hypothesis has been well studied in the past literature (Kaplan et. al. 2020, Allen-Zhu et. al 2019, Frankle and Carbin et. al 2018) and we are not the first ones to state it. Our empirical results with multiplexed smaller models (figure 5) demonstrate multiplexing up to 20 instances, while larger models can multiplex up to 40 instances, which suggests that model capacity might affect multiplexing. Exploring this more empirically (e.g. by using various pruning techniques) is a great direction for future work for better understanding how multiplexing interplays with overparameterization.
>
> **Theoretical construction holding true in practice**: The theoretical construction suggests that under some assumptions, there *exists* an initialization of transformer models, which enables multiplexing. This is only to show that self-attention models are *capable* of multiplexing (as we also mention in the paper). Although our empirical results, especially the one of attention heads not affecting multiplexing much, suggest that our learned models don’t necessarily align with the theoretical construction, the theory provides us an “existence proof” for multiplexing under certain assumptions. We have clarified this better in our revision.
>
> **Accuracy vs Training time plots**: All models in the paper are trained to convergence. We don’t think training longer would improve the performance of these models. Empirically, we see that training time increases sub-linearly with the number of instances. For example, these are number of steps required for convergence on MNLI
>
> | N  | Iterations |
> |----|------------|
> | 1  | 70K        |
> | 2  | 70K        |
> | 5  | 80K        |
> | 10 | 120K       |
> | 20 | 200K       |
> | 40 | 400K       |
>
> **Ablating retrieval warmup training**: Without the retrieval warmup training, we find that the models are unable to learn multiplexing and the task performance is close to random for all tasks.
>
> **Speedups of smaller transformers**:  The goal of these experiments was to show that model size is not a critical factor to enable multiplexing and our DataMUX approach is agnostic to model-size.
>
> **Possibility to learn multiplexing function**: Yes, it is possible! We have presented alternative multiplexing strategies in A.5 and find that learning the multiplexing function by unfreezing the randomly initialized Gaussian vectors for the Hadamard product does not change the performance significantly.

---

> > ### Author Response · Authors · 2022-08-08
> > **Any remaining questions or concerns**
> >
> > Dear reviewer,
> > Thank you again for your review -- please let us know if you have any remaining questions or concerns, so that we can address them before the deadline tomorrow. Alternatively, if you feel that your original concerns are addressed, we would appreciate your updating your evaluation to reflect that.

---

> > > ### Comment · Reviewer_DLX3 · 2022-08-09
> > > **One remaining issue**
> > >
> > > Thank you for your detailed response and improvements to the manuscript. I appreciate that most of my concerns were addressable by clarification.
> > >
> > > My only remaining quibble is that I feel the efficiency/resource angle is still insufficiently addressed, despite it being the main practical motivation for the work. There are a few aspects to this:
> > >
> > > - A 3-20x runtime improvement at the cost of a 2-4% accuracy drop seems like a good value to me, but it would be useful to have alternative inference acceleration techniques to use as meaningful baselines.
> > > - What are the throughput/runtime improvements for multiplexed models that have the same accuracy as non-multiplexed models (i.e. the isoaccuracy regime)?
> > > - How much more expensive does multiplexing make training?
> > >
> > > I would really love to see these questions addressed, even though I will still recommend acceptance if they're not :)
> > >
> > > I have updated my scores as follows:
> > > Presentation: 3->4
> > > Soundness: 2->3
> > > Rating: 6->7

---

### Meta-Review · Area_Chair_nfGt · 2022-08-27

**Recommendation:** Accept
**Confidence:** Certain

**Metareview:**

The paper applies DataMUX  for deep learning approach, which enables the network to process multiple inputs.  The idea is very interesting and the paper is well written.  All three reviews suggested the paper should be acceptable.

**Award:**

No

---

### Decision · Program_Chairs · 2022-09-14

Accept